

**Modeling surface water and groundwater mixing and mixing-dependent denitrification**
**with bedform dynamics**
Xue Ping[1], Zhang Wen[1,2*], Yang Xian[1], Menggui Jin[1,2], Stefan Krause[3]
[1]Hubei Key Laboratory of Yangtze Catchment Environmental Aquatic Science, School of
Environmental Studies, China University of Geosciences, Wuhan, China
[2]State Key Laboratory of Biogeology and Environmental Geology, China University of Geosciences,
Wuhan, China
[3]School of Geography, Earth and Environmental Sciences, University of Birmingham, UK
*Corresponding author: Zhang Wen (wenz@cug.edu.cn)





**Abstract** The hyporheic zone (HZ), where surface water (SW) and groundwater (GW) interact and
mix, acts as a critical interface that attenuates contaminants through enhanced biogeochemical
cycling. While bedform migration significantly influences hyporheic exchange and non-mixing-
driven reactions of solutes from upstream SW, the effects of bedform migration on SW-GW mixing
dynamics and mixing-triggered biogeochemical reactions—particularly under gaining stream
conditions—remain poorly understood. Pioneering a coupled hydrodynamic and reactive transport
model that incorporates bedform migration this paper systematically examines nitrogen processing
for scenarios of variable sediment grain size, stream velocities, and upwelling GW fluxes. Results of
this study reveal that SW-GW mixing and mixing-triggered denitrification zones progressively
transition from crescent shapes into uniform band-like configurations as bedforms migrate. Both
hyporheic exchange flux and mixing flux increase with increasing stream velocity and associated
bedform celerity. The mixing proportion and mixing zone size increase at the start of migration,
while they remain approximately constant when turnover becomes the dominant water exchange
mechanism for fine-medium sandy riverbed. Fast stream flows and migrating bedforms reduce solute
residence timescales and limits denitrification opportunities. Consequently, nitrate removal efficiency
from both stream- and groundwater-borne sources decreases significantly with bedform migration in
fine-medium sandy sediments. The self-purification capacity of the HZ, and particularly its
functioning as a natural barrier against GW contamination, is hindered under such dynamic bedform
conditions. These findings highlight the need to maintain stable bedform conditions in restoration
projects to enhance the capacity of HZ contaminant attenuation.
**1. Introduction**



Anthropogenic activities such as the intensification of agricultural practices with its increased
used of mineral and organic fertilizers, together with high livestock densities and emissions of
inadequately treated domestic and industrial wastewater have significantly increased nitrogen
loading to rivers and groundwater, which impacts water quality, causing eutrophication, hypoxic and
related deterioration of ecosystem functions (Conley et al., 2009; Rouse et al., 1999). Long-term
regulatory monitoring data (e.g., from the UK) indicate that nitrate levels have stabilized in many
rivers, while nitrate concentrations in groundwater-fed rivers continue to increase (Burt et al., 2011;
Howden and Burt, 2008). The persistence of nitrate contamination in groundwater and associated
risks of a "nitrate time bomb" (Ascott et al., 2019) has highlighted the urgency of exploring the
potential of natural microbial processes to mediate nitrate transformation and removal in riverbed
sediments (Shelley et al., 2017; Lansdown et al., 2015; Rivett et al., 2008).
The hyporheic zone (HZ) has received significant attention for its potential to facilitate
enhanced nitrate transformation and removal via denitrification that is a primary process permanently
reducing nitrate, with hyporheic exchange flows (HEFs) acting as a critical mechanism for
transporting nitrate-rich surface water to the riverbed sediments where microbial activities and
biogeochemical reaction rates are enhanced (Boano et al., 2014; Boulton et al., 1998; Cardenas,
2015; Xian et al., 2022; Krause et al., 2022). It has for long been assumed that predominantly stream
waters provide inputs of bioavailable (mainly dissolved) organic carbon (DOC), oxygen ($O_2$) and
nitrate ($NO_3$) into the riverbed where the residence and rection times determine the occurrence of
aerobic respiration and the potential for shifts into anaerobic conditions that may facilitate
denitrification along the HEF paths in the presence of enough remaining DOC (Zarnetske et al.,
2011a, b). These hydrological and biogeochemical mechanisms are in this form mainly

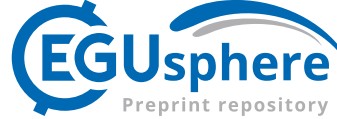

representative of headwater streams, where the HEF is induced by stream flow turbulence (Boano et
al., 2011; Roche et al., 2018, 2019), local geomorphological setting (Cardenas et al., 2008; Marzadri
et al., 2012; Tonina and Buffington, 2007), and flow obstacles such as woody debris, streambed
engineering or restoration structures (Briggs et al., 2013; Wondzell et al., 2009), and hyporheic
nitrate removal processes expected to mainly occur at the middle to end- hyporheic flow paths within
HEF cell sediments.
In lowland systems, groundwater-fed streams and rivers in permeable catchments will be
dominated by base flow of often nitrate enriched groundwaters. The subsurface hydrological
conditions are driven by horizontal HEF as well as vertical fluxes resulting from regional
groundwater flow toward (or from) the stream. Spatial variability in regional groundwater flow can
significantly affect hyporheic exchange and biogeochemical cycling (Boano et al., 2013; Krause et
al., 2013; Munz et al., 2011; Naranjo et al., 2015). It is important to note that the groundwaters of
many piedmont plains and lowlands are often contaminated with high nitrate concentrations, but
usually low in bio-available DOC (Krause et al., 2022). When nitrate-rich groundwater upwells
through deeper sediments and reaches a region enriched in availability of DOC, nitrate reduction
processes are significantly facilitated in the condition (Krause et al., 2009; Naranjo et al., 2015; Ping
et al., 2023; Trauth et al., 2017). Stelzer and Bartsch (2012) developed such a conceptual framework
of nitrate-rich gaining fluvial setting from 8 sites in the Waupaca River Watershed with three order
magnitude in groundwater nitrate concentration. Lansdown et al. (2014) also measured high
denitrification rate at deep sediment in the coarse-grained sediments typical of groundwater-fed
system, located within the River Leith (Cumbria, UK) where diverse nitrogen concentration changes
were confirmed earlier (Krause et al., 2009).





Turnover and removal of the large amounts of $NO_3^-$ from groundwater require DOC either from
autochthonous streambed sources or from downwelling surface water to stimulate nitrate reduction
(Krause et al., 2013, 2022; Ping et al., 2023; Sawyer, 2015; Trauth et al., 2017). For coarse grain or
sandy riverbeds with low autochthonous organic matter content, stream-borne DOC dominates the
supply of carbon sources for nitrogen transformation processes. Sandy sediments with less
autochthonous organic carbon sources covering the majority of alluvial riverbeds are commonly
characterized by topographical features such as ripples, dunes, and riffle-pool sequenced. The typical
and multiple HEF cells induced by bedform topography are generally in crescent shapes (Fox et al.,
2014; Wu et al., 2024). The downward advection of stream borne DOC provides electron donor and
mixes with nitrate-rich and anoxic groundwater. It has been shown that the highest potential for
mixing triggered denitrification is often found at the margin of the HEF cells, which represents the
last natural protection before nitrate enter a stream (Hester et al., 2013, 2014; Gu et al., 2008;
Nogueira et al., 2022).
The effects of mixing triggered denitrification on groundwater borne nitrate transformation in
HZs have been studied almost exclusively for the case of stationary, that is immobile bedforms
(Hester et al., 2017, 2019; Trauth and Fleckenstein et al., 2017; Ping et al., 2023). Bedforms are
mobile in dynamic equilibrium or undergo constant changes during periods of moderate to high
stream flow, and they are typically found in medium and larger waterways under realistic field
conditions (Bartholdy et al., 2015; Risse-Buhl et al., 2023; Schindler et al., 2015). For example,
Harvey et al. (2012) observed the migrating bedforms of dunes (with a median grain size $D_{50}$ of 380
µm) at a velocity of 57.6 cm/h during base flow in the "Clear Run" stream in eastern North Carolina,
USA. Ahmerkamp et al. (2017) found that the ripple bedforms for sands ($D_{50} = 63$ µm) ranged from



11 to 29 cm with a constant ratio of bedform height and length at 1/9, and migrated at velocities of
0.7–6.5 cm/h in the German Bight, Southeastern part of the North Sea. Bedforms migration
complicates the development of hyporheic flow fields, increases solute exchange, alters redox
conditions in riverbeds, and affects contaminant transport and transformation (Ahmerkamp et al.,
2015; Schulz et al., 2023; Peleg et al., 2024). Specifically, bedform migration has negative influences
on non-mixing-dependent denitrification (where nitrate and DOC are both derived from surface
water and travel together along the flow paths) rate and nitrate removal efficiency (Jiang et al., 2022;
Kessler et al., 2015; Ping et al., 2022; Zheng et al., 2019). However, no studies have yet investigated
and explored the effects of bedforms migration on mixing of surface water and groundwater, as well
as its controls and implications for mixing triggered denitrification reaction in groundwater-fed
streams and rivers.

In this study, numerical modeling of hyporheic flow and multi-component solute transport is

used to study the effect of bedform migration on mixing-dependent denitrification in the HZ of a
gaining river, where the overlying water is induced into the sediment by ripple-type bedforms. The
objectives of this study are to determine the effects of bedform migration on the overall extent and
magnitude of mixing of surface water and upwelling groundwater, as well as its influences on
groundwater borne nitrate transport and transformation.
**2. Methods**
**2.1 Model description**

Hyporheic flow, solute transport, and biogeochemical reactions were modelled in saturated

sediments beneath a riverbed. Ripples form in the riverbed and migrate downstream due to sediment





bedload transport processes. The stream geometry is parameterized through its slope $S$, average water
depth $H$, and mean velocity $U$. Triangular shaped ripple bedforms of wavelength $\lambda$ are considered to
develop and migrate downstream by a unidirectional average velocity $u_c$. Flow is driven by pumping
(pressure variation along the bedform surface) and bedform migration processes and influenced by
upwelling groundwater.

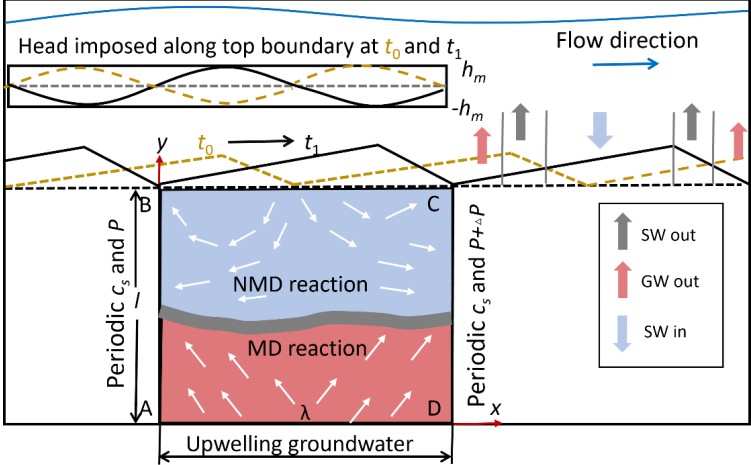


**Figure 1.** Schematic of the model domain with bed form geometry and boundary conditions. Stream
flow and bedform migration are from left to right. The dashed yellow lines represent the streambed
surface and head profile at time $t_0$, and the solid black lines represent the streambed surface and head
profile after migration at $t_1$. "SW in" is where surface water enters the riverbed, and "SW out" is
where surface water discharges to the stream, and "GW out" is along the upstream and downstream
sides of bedforms where groundwater discharges to the stream. The gray band represents the mixing
zone of surface water and groundwater. NMD reaction=non-mixing-dependent reaction and MD
reaction=mixing-dependent reaction.

As bed migration occurs, sediment erosion and deposition primarily take place in the surface



layer of the streambed, thereby leading to the formation of bedforms such as ripples and dunes. In
contrast, sediments in the deeper streambed layers remain relatively stable (Harvey et al., 2012;
Precht et al., 2004). As illustrated in Figure 1, from the perspective of an observer stationary at a
target riverbed segment with horizontal length $\lambda$, bedforms and their associated pressure fields
migrate downstream at velocity $u_c$ over the deeper immobile sediments (Ping et al., 2022; Teitelbaum
et al., 2022). Given that ripple heights are negligible compared to the depth of the stable sediment
domain, the undulating riverbed can be reasonably approximated as a flat bed with time-varying
pressure patterns during bedform migration. Given the periodic nature of the morphological feature,
we only focused on a representative section and constructed a two-dimensional rectangular domain
of length 0.2 m and depth 0.16 m for modeling.
**2.2 Model formulation**

The pore water flow was calculated using Darcy's law and the groundwater flow equation:

$$\nabla \cdot \left( -K \nabla h \right) = 0 \tag{1}$$

where $h$ [L] is the hydraulic head, and $K$ [L T$^{-1}$] is the hydraulic conductivity.

The head profile on the streambed surface (BC) was described as a sinusoidal function that

moves downstream by the ripple's migration velocity $u_c$ (Ping et al., 2022):
$$h\big|_{y=l} = h_m \cdot \sin m(x - u_c dt) \tag{2}$$

where $h_m$ [L] is the amplitude of the head variation, $m$ [-] is the wave number of the variation ($m =$
$2\pi/\lambda$), the head difference is related to the properties of the overlying flow (Elliott and Brooks, 1997):
$$h_m = a \frac{U^2}{2g} \left( \frac{H_d / H}{0.34} \right)^n \tag{3}$$



where $a = 0.28$ [-] is a dimensionless coefficient, $U$ [L T$^{-1}$] is the average stream velocity, $H_d$ [L] is
the height of the ripple, $H$ [L] is the water depth, and $g$ [L T$^{-2}$] is the gravity acceleration. The
exponent $n$ equals to 3/8 if $H_d < 0.34H$ and 3/2 otherwise.

The transport of reactive solutes within the streambed sediment was described by the advection-

dispersion-reaction equation:

$$\frac{\partial c_i}{\partial t} - \nabla \cdot \left( \boldsymbol{D_{ij}} \nabla c_i \right) + \nabla \boldsymbol{v} \cdot c_i = R_i$$

(4)

where $c_i$ [M L$^{-3}$] represents the concentration of reactive components, $\boldsymbol{v}$ [L T$^{-1}$] is the seepage or
linear pore water velocity, $\boldsymbol{D_{ij}}$ [L$^2$ T$^{-1}$] is the hydrodynamic dispersion and is defined by Bear and
Verruijt (1998):

$$\boldsymbol{D_{ij}} = \left( \alpha_L - \alpha_T \right) \cdot \frac{v_i v_j}{|v|} + \delta_{ij} \cdot \left( \alpha_T |v| + \theta \cdot \iota D_m \right)$$

(5)

where $\alpha_L$ [L] and $\alpha_T$ [L] are longitudinal and transverse dispersivities, respectively, $D_m$ [L$^2$ T$^{-1}$] is
molecular diffusion coefficient and $\iota$ [-] is tortuosity.

The reactive transport model considered three chemical species: DOC, $O_2$, and $NO_3^-$. In order to

distinguish non-mixing-dependent and mixing-dependent denitrification, we divided nitrate into two
separate pools, denoted s-$NO_3^-$ for nitrate transported from the surface water and g-$NO_3^-$ for nitrate
from upwelling groundwater. The biogeochemical reactions are aerobic respiration (AR), non-
mixing-dependent, and mixing-dependent denitrification (DN):

$$AR : CH_2O + O_2 \rightarrow CO_2 + H_2O$$                                       (R1)

$$DN : CH_2O + 0.8NO_3^- + 0.8H^+ \rightarrow CO_2 + 1.4H_2O + 0.4N_2$$                    (R2)



Reaction rates were defined using Monod kinetics (Zarnetske et al., 2012):

$$R_{AR} = V_{AR} \times X_{AR} \times \frac{c_{O_2}}{c_{O_2} + K_{O_2}} \times \frac{c_{DOC}}{c_{DOC} + K_{DOC}}$$

(6)

$$R_{DN} = V_{DN} \times X_{DN} \times \frac{c_{s/g-NO_3^-}}{c_{s/g-NO_3^-} + K_{NO_3^-}} \times \frac{c_{DOC}}{c_{DOC} + K_{DOC}} \frac{K_{inh}}{K_{inh} + c_{O_2}}$$

(7)

The reaction terms $R_i$ was given by

$$R_{s-NO_3^-} = -R_{DN}(c_{s-NO_3^-})$$

(8)

$$R_{g-NO_3^-} = -R_{DN}(c_{g-NO_3^-})$$

(9)

$$R_{O_2} = -R_{AR}$$

(10)

$$R_{DOC} = -R_{AR} - 0.8(R_{DN}(c_{s-NO_3^-}) + R_{DN}(c_{g-NO_3^-}))$$

(11)

where $V_{AR}$ and $V_{DN}$ [T$^{-1}$] are the maximum reaction rate of aerobic respiration and denitrification, $X_{AR}$
and $X_{DN}$ [M L$^{-3}$] are the biomass of functional microbial groups facilitating the reaction components
of AR and DN. $K_{inh}$ [M L$^{-3}$] is a non-competitive inhibition factor used for representing inhibition of
DN given oxygen availability.
The lateral boundaries AB and DC were set to be periodic boundaries: $P(0, y, t) = P(\lambda, y, t)$
$+\Delta P$, $c_i(0, y, t) = c_i(\lambda, y, t)$ and $\partial c_i(0, y, t)/\partial y = \partial c_i(\lambda, y, t)/\partial y$. The additional pressure drop $\Delta P$ [M
L$^{-1}$T$^{-2}$] is derived from the streambed gradient and calculated using $\Delta P = S\rho g\lambda$. The top boundary BC
was specified as an open Dirichlet boundary with constant solute concentrations in the stream. An
upward groundwater flux with constant nitrate concentration was specified at the bottom boundary
AD to mimic a nitrate polluted groundwater discharge.



**2.3 Bedform migration**

The mechanisms of bedform initialization, formation and migration are initiated through a set of criteria to ensure that ripples are expected to form under the modeled scenarios and reach a state of dynamic equilibrium, where the ripples remain mobile while maintaining their shape. The development of ripples is assessed under different conditions of median particle size $D_{50}$ and flow velocity based on a set of quantitative criteria (as reference in Ping et al., 2022 for criteria on ripple formation). For a specific grain size of streambed sediment, particular stream velocities that fulfill all these criteria for the development of ripple bedforms that are mobile under dynamic equilibrium are selected (Figure 2). All simulation scenarios were listed in the Table S3 in the supplementary material.

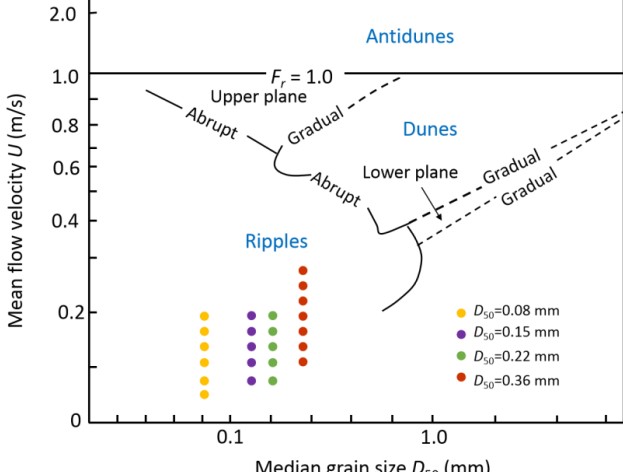

**Figure 2.** The bedform stability diagram (modified from Ashley, 1990) showing the bedform properties and hydraulic conditions considered in this study.

Ripple migration velocities are implemented using an empirical relation after Coleman and Melville (1994), which was derived from flume experiments:

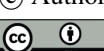



$$\frac{u_c}{\left(u^* - u_{cr}^*\right)\left(\tau^* - \tau_{cr}^*\right)}\left(H_d/D_{50} - 3.5\right)^{1.3} = 40$$


(12)

where $D_{50}$ [L] is the median grain size, $u^* = (gHS)^{0.5}$ [L T$^{-1}$] is the bed shear velocity, $S$ [-] is the
stream gradient and calculated by Chezy equation ($U = H^{2/3}S^{1/2}/n$, where $n$ [-] is the Manning
coefficient and assumed to be 0.02 for sand). $u_{cr}^*$ [L T$^{-1}$] is the critical bed shear velocity and it can
be calculated by the critical Shield parameter $\tau_{cr}^*$ ($\tau_{cr}^* = \tau_{cr}/g(\rho_s - \rho)D_{50}$, $u_{cr}^* = (\tau_{cr}/\rho)^{0.5}$, $\rho_s$ [M L$^{-3}$]
and $\rho$ [M L$^{-3}$] are the density of sediment and water), and $\tau^*$ is the shield number related to the bed
shear velocity ($\tau^* = u^{*2}/rgD_{50}$). The critical shields parameter defines the threshold for the
initialization of motion. The derived celerity was substituted into Equation (2) to determine the
migration of sinusoidal head profile with ripples moving.
**2.4 Governing non-dimensional numbers**
The characteristics of the modeled system were depicted by a series of non-dimensional
numbers, which represent the relative dominance of various forces, transport, and reaction processes
in this system. Firstly, we used the Reynolds number $Re$ to characterize the flow condition of surface
water (Cardenas and Wilson, 2006):

$$Re = \frac{UH_d}{\upsilon}$$


(13)

where $\upsilon$ [L$^2$ T$^{-1}$] represents kinematic viscosity of water.
We introduced the dimensionless parameter $U_r$ [-] to represent the relative magnitude of
bedform celerity and the pressure-induced pore water velocity driven by pressure variation over the
ripple surface and upwelling groundwater:





$$U_r = \frac{\theta \cdot u_c - u_s}{u_p}$$

$\qquad$ (14)
where $u_s$ [L T$^{-1}$] is the seepage velocity of the underflow induced by stream gradient ($u_s = KS$), and
thus the characteristic horizontal velocity is $u_c - u_s/\theta$. $u_p/\theta$ [L T$^{-1}$] is the pore water velocity induced
by pumping process and is calculated using the analytical solution after Boano et al. (2009) and Fox
et al. (2014) accounting for vertical groundwater flux ($u_q$):

$$u_p = u_{p,0}\sqrt{1 - \left(u_q / \pi u_{p,0}\right)^2} + \left(\left|u_q\right| / \pi\right)\sin^{-1}\left(\left|u_q\right| / \pi u_{p,0}\right) - \left(\left|u_q\right| / 2\right)$$

$\qquad$ (15)

$$u_{p,0} = a\frac{KU^2}{g\lambda}\left(\frac{H_d/H}{0.34}\right)^m$$

$\qquad$ (16)
where $u_{p,0}$ represents the hyporheic exchange driven by pressure variation over the sediment-water
interface; if $U_r > 1$, turnover process dominates and controls the hyporheic exchange, otherwise, the
system is pumping process dominated (Jiang et al., 2022).
$\qquad$ The relative magnitude of hyporheic exchange flux driven by pressure variation along the ripple
surface and upwelling groundwater flux is determined as:

$$U_b = \frac{u_q}{u_{p,0}}$$

$\qquad$ (17)
$\qquad$ The relative dominance of hyporheic exchange and biogeochemical reaction in nitrate removal
can be defined by the Damköhler number (Ocampo et al., 2006; Zarnetske et al., 2012; Zheng et al.,

2019):

$$Da = \frac{\tau_T}{\tau_R}$$

$\qquad$ (18)
the characteristic timescale for the transport of solutes through the ripple is estimated as (Azizian et





al., 2015):
$$\tau_T = \frac{\lambda\theta}{\pi^2 u_p} \qquad (19)$$

and the reaction timescale $\tau_R$ represents the time needed to consume dissolved oxygen of hyporheic
water to a prescribed anoxic environment threshold (2 mg/L). The reaction timescale is described as:
$$\tau_R = \frac{\ln\left(c_{O_2}/c_{O_2,\lim}\right)}{V_{AR}} \qquad (20)$$

biogeochemical reactions are transport-limited when $Da < 1$. The biogeochemical reactions
depended on reaction kinetics due to the brevity of the time that reactants spend within the HZ.
Under these low $Da$ conditions, the HZ remains oxic conditions, resulting in a minimal or no
denitrification to occur. Conversely, when $Da > 1$, the residence time of reactants exceeds the
reaction time, and thus oxygen is consumed and favors for the occurrence of denitrification in anoxic
conditions (Jiang et al., 2022; Zarnetske et al., 2011a).
**2.5 Model setup and parametrization**

All parameter values in this study were shown in Table 1. The bedform geometry of Ping et al.

(2022) was adopted for this study (streambed length and height: $\lambda = 0.2$ m and $l = 0.16$ m; the ripples
located at $\lambda_c = 0.15$ m with a height of $H_d = 0.02$ m). Here grain sizes $D_{50}$ of 0.08, 0.15, 0.22 and 0.36
mm were considered, typically falling within characteristic grain diameters on sandy riverbeds
(Ahmerkamp et al., 2017; Harvey et al., 2012). We used the empirical relation $k = Da \times 735 \times 10^6 \times D_{50}{}^2$
(where $Da = 9.869 \times 10^{-13}$ is the conversion factor for unit Darcy to m$^2$; Gangi, 1985).

The concentrations of DOC, $O_2$, and s-$NO_3^-$ in stream were specified as 30 mg/L, 8 mg/L, and 5

mg/L. This configuration represents a pristine stream characterized by moderate nutrient levels



(Ocampo et al., 2006). The g-$NO_3^-$ in groundwater was set as 15 mg/L, representing the chemical
signature of nitrate-contaminated groundwater that lacks both oxygen and organic matter (Hester et
al., 2014). The maximum reaction rate and corresponding functional microbial concentration for AR
and DN are listed in Table 1, the choose biogeochemical values are consistent with the parameter
setting of nutrient cycling in hyporheic zones and riparian zones (Gu et al., 2008; Nogueira et al.,
2021; Zarnetske et al., 2012).
**Table 1.** Model parameters used in numerical simulations

| Parameter | Description | Value |
|---|---|---|
| $l$ [m] | Streambed depth | 0.16 [a, b] |
| $l_c$ [m] | Ripple crest | 0.15 [a, b] |
| $\lambda$ [m] | Wavelength of ripple | 0.2 [a, b] |
| $H_d$ [m] | Height of ripple | 0.02 [a, b] |
| $H$ [m] | Stream water depth | 0.1 [a, b] |
| $\theta$ [1] | Porosity | 0.38 |
| $\alpha_L$ [m] | Longitudinal dispersivity | 0.01 |
| $\alpha_T$ [m] | Transverse dispersivity | 0.001 |
| $K_{inh}$ [mg L$^{-1}$] | Inhibition constant | 0.25 [c, d] |
| $K_{DOC}$ [mg L$^{-1}$] | Half-saturation constant for dissolved organic carbon | 6 [c, d] |
| $K_{NO3}$ [mg L$^{-1}$] | Half-saturation constant for nitrate | 1 [c, d] |
| $K_{O2}$ [mg L$^{-1}$] | Half-saturation constant for dissolved oxygen | 0.5 |
| $V_{DN}$ [h$^{-1}$] | Maximum specific uptake rate for denitrification | 1 [c, d] |
| $V_{AR}$ [h$^{-1}$] | Maximum specific uptake rate for aerobic respiration | 2 [c, d] |





| $C_{DOC}$ [mg L$^{-1}$] | Concentration of dissolved organic carbon in stream | 30 |
|---|---|---|
| $C_{O2}$ [mg L$^{-1}$] | Concentration of dissolved oxygen in stream | 8 |
| $C_{s\text{-}NO3^-}$ [mg L$^{-1}$] | Concentration of nitrate in stream | 5 |
| $C_{g\text{-}NO3^-}$ [mg L$^{-1}$] | Concentration of nitrate in groundwater | 15 |

[a] Janssen et al. (2012) [b] Ping et al. (2022) [c] Zarnetske et al. (2012) [d] Sawyer (2015)
The following distinct model experiments were carried out: The Reynolds number, i.e., mean
stream velocity, was varied for $Re$ = 2000–6000 in intervals of 500, with corresponding stream water
velocities of $U$ = 0.1–0.3 m/s. The range of $U_b$ was set from 0.3 to 0.7 in intervals of 0.1. A larger
upward flux than $0.9 \times u_{p,0}$ would eliminate the entire hyporheic flow cell, thus we set the maximum
boundary flux at a value slightly below this threshold. Meanwhile a minimum of $0.2 \times u_{p,0}$ ensures
that upwelling groundwater is still mixing with surface water with minor influences on hyporheic
flow cell.
The finite element software, COMSOL Multiphysics (version 6.1) was used to solve the Darcy
flow and multi-component solute transport model. The domain was discretized with a grid spacing
from $4 \times 10^{-4}$ to 0.2 cm, the resultant mesh consisting of 19,940 elements. To maintain a constant
bedform displacement ($\Delta x$) per timestep, the simulation was conducted with $\Delta x$ = 2 cm, while $dt$ was
adjusted in inverse proportion to the migration celerity $u_c$. The total duration of the simulation was
set to be equal to the time needed for hundreds of ripples to travel across the modeled domain until
the hyporheic exchange and biogeochemical processes reached quasi-steady states.
**2.6 Model Metrics**
**2.6.1 Mixing of surface water and groundwater**

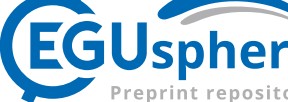

Mixing occurs at the interface between regions of advected surface water and upwelling
groundwater, where the flow paths from these two sources run parallel to each other. We specified a
constant concentration of conservative tracer ($c_{gw}$ = 1 mg/L) in groundwater to represent
groundwater. The tracer plume is used to visualize the surface water and groundwater mixing size
and calculate the mixing magnitude quantitatively. The mixing zone is defined as the area where the
groundwater proportion varies between 16% and 84% (Santizo et al., 2020). The size of the surface
water and groundwater mixing zone ($A_{mix}$) was calculated by integrating the area where the
concentration of the conservative tracer ranges from 0.16 to 0.84 mg/L.
The amount of tracer mass that undergoes mixing as it transitions from flow paths originating at
the bottom boundary to those emerging at the streambed surface was quantified to determine the net
effect of mixing occurring along the entire length of the mixing zone. The streambed surface was
divided into three zones (Figure 1): "SW IN", where surface water enters the bed; "SW OUT", where
surface water discharges back into the overlying water column; and "GW OUT", where upwelling
groundwater discharges into the stream. The mass flux for the SW OUT zone was used to describe
mixing. If no mixing occurred, all the conservative tracer mass entering the model at the bottom
boundary would exit through the GW OUT zone on the streambed surface. The mixing flux (MF)
across the sediment-water interface was determined by integrating the Darcy flux flowing outward
through the SW OUT zone, representing the total cumulative effect of mixing along the entire length
of the mixing zone (Hester et al., 2013).
**2.6.2 Nitrate reaction rate and efficiency**
When the hydro-physical and biogeochemical conditions reach a quasi-steady state, we select



the last 10 periods of ripple migration and calculate the total amount of stream- or groundwater-
borne nitrate being induced into the riverbed during the time interval:
$$M_{in,s-NO_3^-} = B\int_{\Delta T} c_{s-NO_3^-} F_{SWI}dL_{top}dT \tag{21}$$

$$M_{in,g-NO_3^-} = B\int_{\Delta T} c_{g-NO_3^-} qdL_{bottom}dT \tag{22}$$

where $B$ [L] is the per unit width, $F_{SWI}$ [L T$^{-1}$] is the inward flux across the ripple surface.

The total amount of nitrate removed by non-mixing-dependent (NMD) denitrification and

mixing-dependent (MD) denitrification are calculated as follows during the same time interval:
$$M_{NMD} = B\int_{\Delta T}\int R_{DN}(c_{s-NO_3^-})dAdT \tag{23}$$

$$M_{MD} = B\int_{\Delta T}\int R_{DN}(c_{g-NO_3^-})dAdT \tag{24}$$

where $A$ [L$^2$] is the area of the streambed.

The nitrate removal efficiency is defined as the ratio of the amount of nitrate being removed by

DN to the amount of nitrate being induced into the riverbed:
$$N_{RE-NMD} = \frac{M_{NMD}}{M_{in,s-NO_3^-}}, \quad N_{RE-MD} = \frac{M_{MD}}{M_{in,g-NO_3^-}} \tag{25}$$

**3. Results**
**3.1 Model validation**

The model development was validated by comparison to flume experiments of Wolke et al.

(2019) that were conducted to study the evolution of oxygen in the riverbed under different
conditions of mean stream velocity (0.16−0.32 m/s) and bedform migration celerity (0−0.394 cm/h).
No upwelling flux of groundwater was considered at the bottom of the riverbed. The experiment was

designed with a total of 5 operating conditions, each of which was repeated twice and labeled as Set

1 and Set 2. Based on the criteria for ripple migration, it was determined that under the hydraulic

conditions of Run 5, ripples could not migrate while maintaining their shapes due to increased flow

intensity. Therefore, the model validation simulations considered four hydraulic conditions of Run 1

to 4.

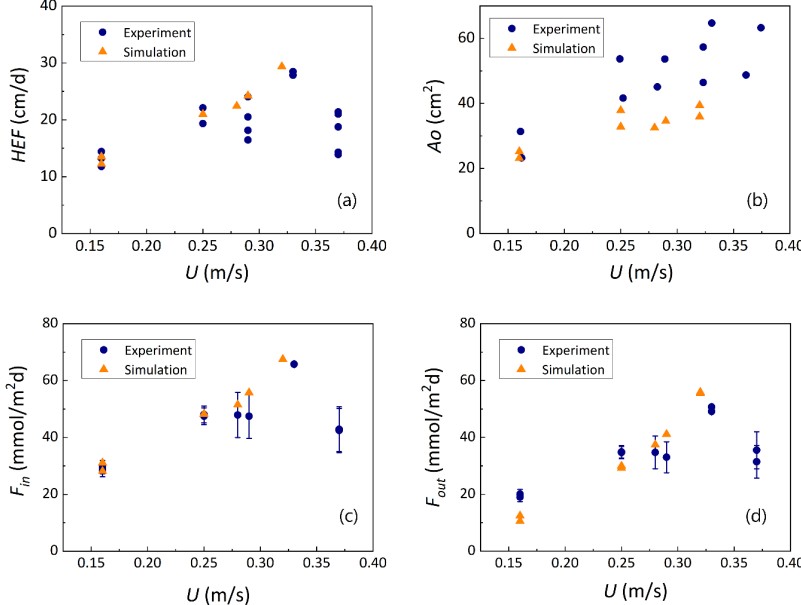

**Figure 3.** Comparison of numerical modeled (a) hyporheic exchange flux, (b) summed oxic zone and

(c) oxygen influx and (d) oxygoutflux and experimental measurements by Wolke et al. (2019) under

various conditions.

The parameters used for model validation are shown in supporting information Table S1 and S2.

In stationary and slow-migrated beds, the spatial distribution of oxygen creates a typical

conchoidally-shaped plume in the riverbed. In contrast, for fast-migrated beds, the oxygen plume

becomes a more uniform front (Ping et al., 2022). The comparison of modeled oxygen distributions

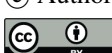



with experimental measurements reveals that simulated values of the oxygenated zone was slightly
lower than observed. This discrepancy is mainly attributed to two factors: firstly, the dissolved
oxygen concentrations measured by the planar optode system were relatively high, as noted in Wolke
et al. (2019) themselves. Secondly, only oxygen fluxes within immobile riverbeds were simulated
and did not include the areas of the mobile sections in the numerical modeling. Because the mobile
riverbeds exist in oxygen-rich environments, this exclusion led to the simulated values being lower
than the measured values. Overall, the simulated hyporheic exchange flux, oxygen area and oxygen
fluxes are displayed in Figure 3 and they are in good agreement with the measurements of Wolke et
al. (2019) in trend, suggesting that the mobile bedform model is capable to reproduce realistic
conditions well.
**3.2 Effect of bedform migration on mixing regimes and solute transport**

To simulate the range of natural environmental conditions, the reactive transport equations were

solved for different stream velocities, grain sizes and groundwater upwelling fluxes, which include a
corresponding range of ripple migration celerites and sediment permeabilities derived from the
empirical relations. As an example, the patterns of pore water transport as well as SW and GW
mixing are shown for a grain size of 0.15 mm, a constant ratio between pumping driven hyporheic
exchange flux and upwelling GW flux $U_b = 0.6$ and four different stream velocities, that is four
different $Re$ numbers.

For low surface water flow velocity ($Re = 2500$, $U = 0.125$ m/s), no migration of bedform was

predicted by the model. SW enters the sediment in the high-pressure region on the stoss side, flows
through the porous medium, and exits the bedform in the low-pressure region on the lee side,
forming a typical conchoidally shaped hyporheic flow cell. Upwelling GW is diverted around the



hyporheic flow cell, mixes with SW, and then exits into the overlying water from both sides, in
patterns similar to those shown previously by Fox et al. (2014) and Hester et al., (2019). SW and GW
mixing zone (i.e. the mixing area where the fraction of GW ranges between 16% and 84%) emerges
as a thin band along the hyporheic flow cell, and covers over 8 % of the modeled domain.
Meanwhile, NMD denitrification occurs below the oxygen plume with the reactive zone in a
conchoidal shaped distribution, while MD denitrification reactive zone develops along the edge of
the mixing zone, where DOC from SW meets nitrate from GW (see row 1 in Figure 4).

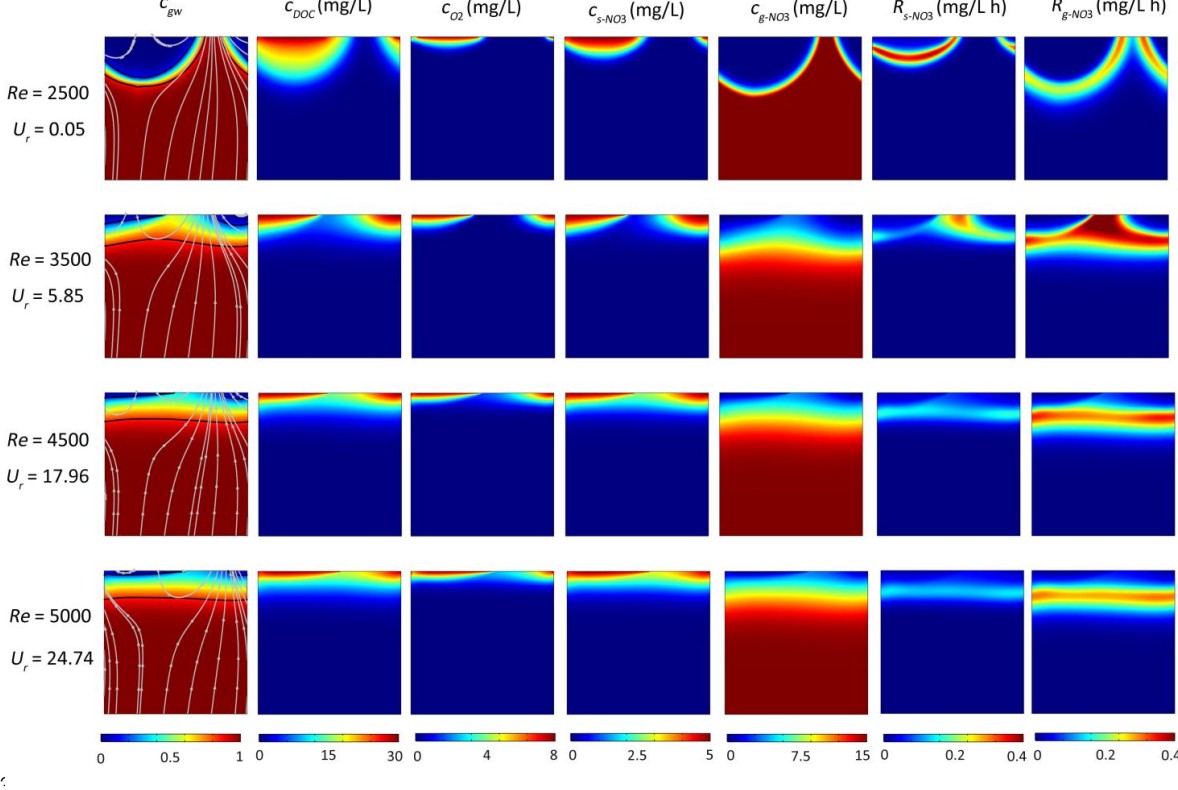

**Figure 4.** Effect of bedform migration on riverbed biogeochemistry for $U_b$ = 0.6 and $D_{50}$ = 0.15 mm.
Shown are profiles of (column 1) conservation solute representing groundwater fraction ($c_{gw}$),





(column 2) DOC concentration ($c_{DOC}$), (column 3) O$_2$ concentration ($c_{O2}$), (column 4) s-NO$_3^-$
concentration ($c_{\text{s-NO3-}}$), (column 5) g-NO$_3^-$ concentration ($c_{\text{g-NO3-}}$), (column 6) non-mixing-
dependent (NMD) denitrification rate ($R_{\text{s-NO3-}}$) and (column 7) mixing-dependent (MD)
denitrification rate ($R_{\text{g-NO3-}}$).
As stream flow velocity increases ($Re = 3500$, $U = 0.175$ m/s), this changes the pressure
distribution patterns with its zones of high and low pressure. Consequently, the simulated hyporheic
flow cells move downstream, while simultaneously also shrinking in size. The shape of the SW and
GW mixing zone changes distinctly, forming a horizontal band with a wider range 17.01% of the
while domain. The penetration of stream-derived solutes into the streambed is reduced, with a more
gradual concentration gradient in the horizontal and vertical directions, whereas the g-NO$_3^-$ plume is
uniformly distributed horizontally. Both NMD and MD denitrification hot spots form in the central
position near the sediment-water interface as the bedform surface (see row 2 in Figure 4).
When bedform migration is further increased ($U_r = 17.96$ and $24.74$), the bedform migration
fully dominates over the pore water flow, and hence, continuous solute layers are found in the
subsurface (as depicted in row 3 and 4 of Figure 4). The penetration depths of stream-borne solutes
are decreased in comparison to those in slow- to medium- fast migrating bedforms. The NMD and
MD denitrification zones become thin and move upward with decreased reaction rates. Similar to the
conclusions obtained in previous studies (Kessler et al., 2015; Zheng et al., 2019), the migration of
the bedform reduces the penetration depth of solute and the scope of hyporheic exchange cell. We
also found that a larger migration celerity increases the size of mixing zone between surface water
and groundwater. The SW and GW mixing zone accounts for 17.74% and 17.86% of the domain
area, respectively.



**3.3 Effect of migration celerity on mixing regimes**

400   The mixing intensity across the bedform surface and the size of the mixing area within the

subsurface are estimated by simulating four different grain sizes and five upwelling groundwater
fluxes, under varying stream velocities associated with the corresponding bedform celerity. The
evolutions of the hyporheic exchange flux, net mixing flux and size of mixing zone with increasing
*Re* number are summarized in Figure 5. In the third row of Figure 5, *Fmix* represents the ratio of the
net mixing flux to the total hyporheic exchange flux. This total flux is triggered by both pumping and
bedform migration, and is simultaneously influenced by upwelling groundwater. *Fmix* serves as a
metric to quantify the proportion accounted for by the surface water-groundwater mixing flux within
the overall flux of surface water and groundwater interaction across the sediment-water interface.

409   As shown in Figure 5, the hyporheic exchange flux increases as the stream velocity and

bedform celerity rise across various grain sizes of the bedform, meanwhile the mixing flux across the
sediment-water interface also increases with the increasing stream velocity, except for some special
circumstances. For riverbed consists of very fine and fine sand ($D_{50} = 0.08-0.15$ mm) under
moderate groundwater discharging conditions ($U_b < 0.5$), the mixing flux increases significantly at
the start of bedform migration, and then the mixing flux across the sediment-water interface begins
to decline with increasing celerity. In fact, this is because higher migration velocities of a riverbed
with relatively low permeability limits the discharge of groundwater into the river (a horizontal
distribution of stream borne solute plume), leading to the mixing of SW and GW primarily occurring
within the streambed (see the column 1 in Figure 4). Concurrently, only the mixing flux through the
water-sediment interface is reduced at this time.

420   For medium sand ($D_{50} = 0.22-0.36$ mm), we found that the SW and GW mixing flux





demonstrates a substantial increase relative to stationary bedforms when bedform migration initiates
at moderate velocities. As migration celerity accelerates, the magnitude of mixing flux gradually
approaches a plateau, with only marginal reductions observed at higher migration celerities. Besides,
the size of SW and GW mixing zone also exhibit the similar trends.

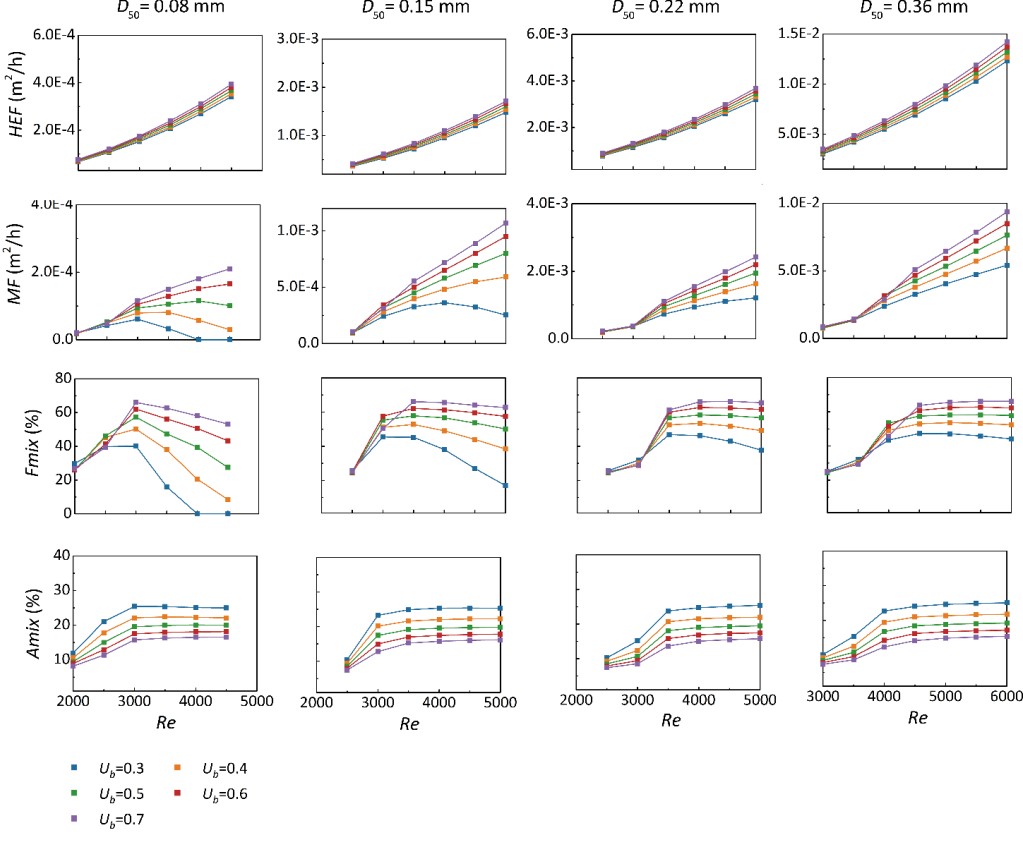


**Figure 5.** The variation of hyporheic exchange flux (*HEF*, row 1) and mixing flux (MF, row 2)
across the sediment-water interface, the proportion of mixing flux to hyporheic exchange flux (*Fmix*,
row 3) and size of mixing zone (*Amix*, row 4) with stream velocity and associated bedform celerity
across different medium grain size when $U_b = 0.3-0.7$.





Overall, bedform migration controls the shape and size of the SW and GW mixing zone,

enhances the magnitude of hyporheic exchange flux and mixing flux. The mixing flux and size is
also influenced by the upwelling GW flux. As $U_b$ increases from 0.3 to 0.6, the mixing flux and the
proportion of mixing flux to total hyporheic exchange flux rise significantly. An increase in $U_b$
reduces the size of the mixing zone because both the hyporheic exchange flow cell and the mixing
zone are confined to shallower depths within the riverbed due to the larger upward flow.
**3.4 Impact of ripples migration on nitrate removal**

To assess the impact of ripple migration on the removal of s-$NO_3^-$ and g-$NO_3^-$ within domains

of varying medium grain sizes, the influx of nitrate into the riverbed and the total reaction rate were
determined.

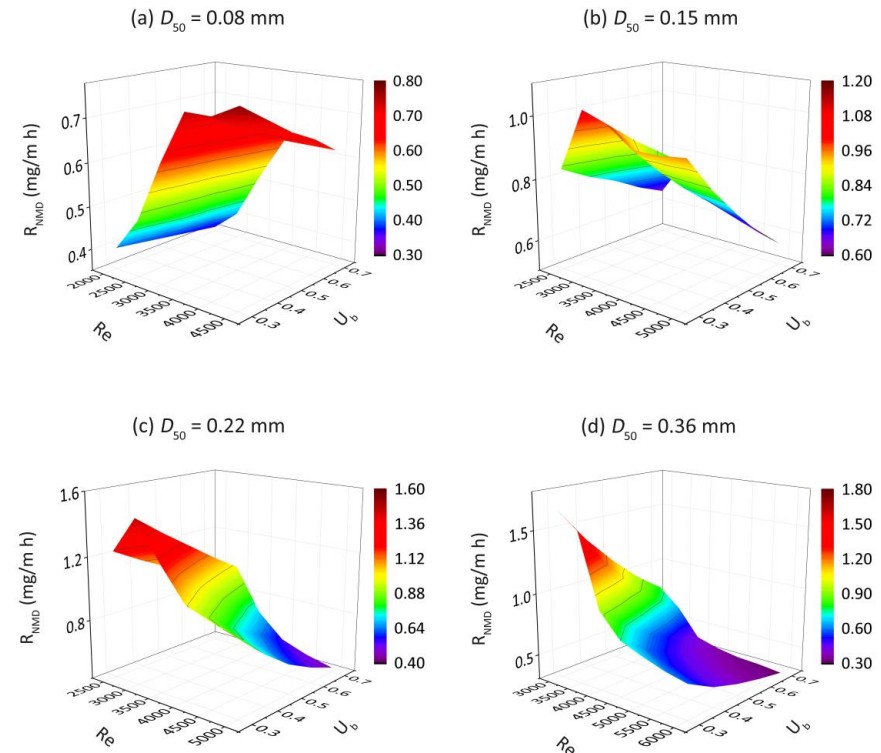


**Figure 6.** The non-mixing-dependent denitrification rates ($R_{NMD}$) as functions of $U_b$ and $Re$ for



different medium grain sizes.

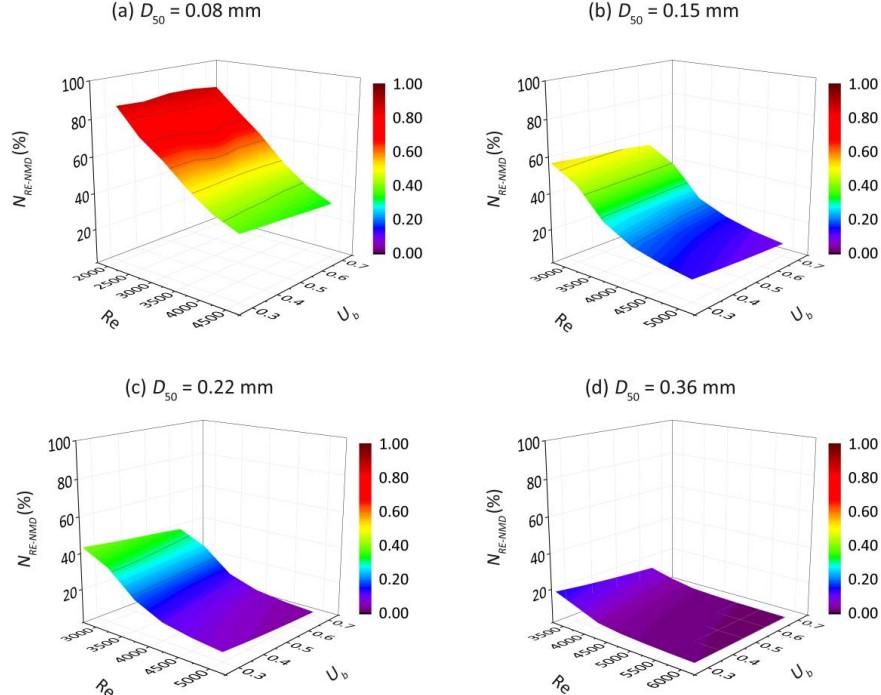

**Figure 7.** The removal efficiencies of stream borne nitrate ($N_{RE-NMD}$) as functions of $U_b$ and $Re$ for
different medium grain sizes.

For s-NO$_3^-$, the NMD denitrification rate increases with both the overlying water velocity and

the migration celerity in very fine sand ($D_{50}$ = 0.08 mm). This is likely caused by higher flow
velocities driving longer advective flow paths and increase solute residence times within the
sediment, thereby enhancing denitrification in reaction-limited systems ($Da$ > 2.85). In contrast, in
riverbeds of fine to medium sand ($D_{50}$ = 0.15−0.36 mm) with higher permeabilities, the reduction
rate of s-NO$_3^-$ is negatively correlated with the mean stream velocity when the system becomes
transport-limited ($Da$ < 2). This is likely because nitrate travels fast along flow paths and does not
undergo denitrification within the moving bedforms (Figure 6). Additionally, the migrating bedforms





enhance the delivery of s-NO$_3^-$ into the sediment due to increased hyporheic exchange flux.
Consequently, the removal efficiency of s-NO$_3^-$ decreases monotonically across various medium
grain sizes (Figure 7).

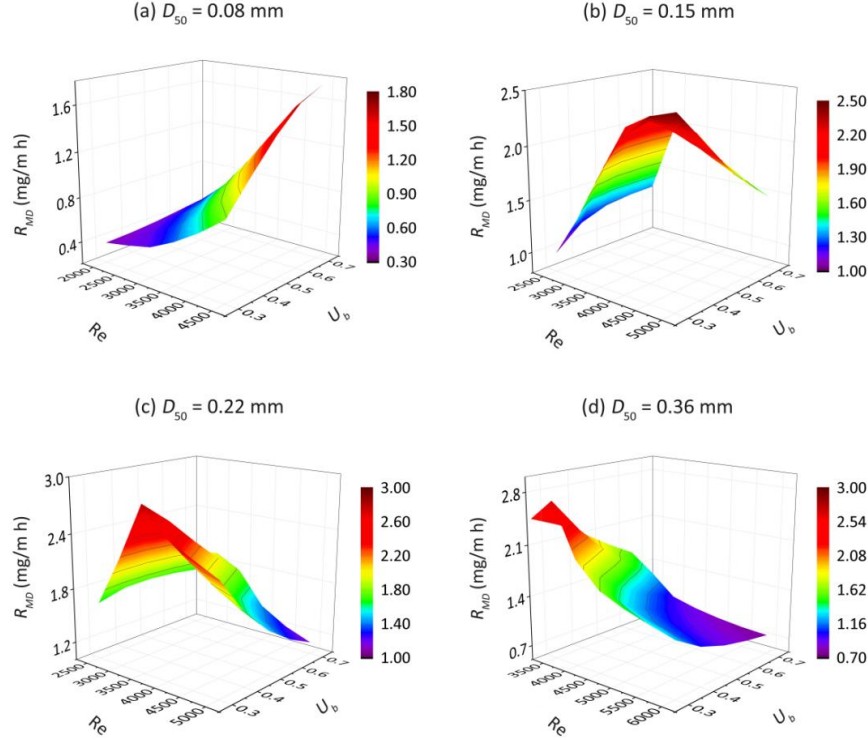

**Figure 8.** The mixing-dependent denitrification rates ($R_{MD}$) as functions of $U_b$ and $Re$ for different
medium grain sizes.

For g-NO$_3^-$, the increase in MD denitrification is also seen for g-NO$_3^-$ at low to medium $Re$

when $D_{50} < 0.36$ mm (Figure 8). Compared to s-NO$_3^-$, the advective flow paths and residence times
of g-NO$_3^-$ in groundwater are longer. Consequently, the reduction rate of g-NO$_3^-$ decreases only in
riverbeds consisting of medium sand with fast bedform migration celerity ($Re > 4000$), as the solute
residence time is significantly reduced. Interestingly, the rise in the MD denitrification rate
compensates for the increased nitrate influx in very fine sand ($D_{50}$ = 0.08 mm) at high stream
velocity. Most of g-$NO_3^-$ that enters the sediment is consumed before entering the overlying water
column. For fine to medium sand riverbed ($D_{50}$ = 0.15–0.36 mm), the g-$NO_3^-$ removal efficiency
decreases strongly with increasing $Re$. The natural protective role of the SW and GW mixing zone in
preventing nitrate-contaminated groundwater from entering rivers is being hindered (Figure 9).

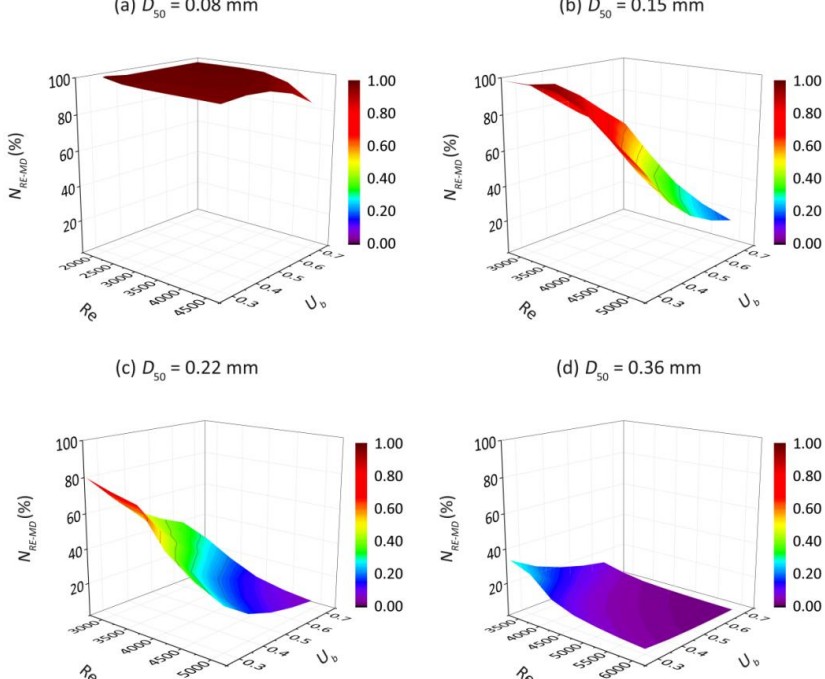


**Figure 9.** The removal efficiencies of groundwater borne nitrate ($N_{RE-MD}$) as functions of $U_b$ and $Re$
for different medium grain sizes.
**4. Discussion**

This study for the first time quantified the effect of bedform migration on surface water and

groundwater mixing process as well as mixing triggered denitrification. Previous research has
primarily focused on the potential impacts of bedform migration on hyporheic exchange driven by
streambed morphological features, as well as non-mixing-dependent biogeochemical processes



where reactants are assumed to reside predominantly in surface water. However, such studies
represent only a small subset of possible streambed environmental conditions, focusing exclusively
on specific headwater stream conditions (Jiang et al., 2022; Kessler et al., 2015; Ping et al., 2022;
Zheng et al., 2019). The impact of bedform migration on the conceptual model of bedform-induced
hyporheic exchange, which is influenced by groundwater upwelling and/or ambient lateral
groundwater flow in the mid-stream section of lowland rivers, has received relatively less attention
and examination.

In streams and rivers that are fed by regional groundwater and possess undulating bedforms,

surface water gets mixed with groundwater throughout the local hyporheic exchange process. The
mixing zone exhibits a typical crescent shape along the periphery of typical hyporheic exchange cells
within a stationary streambed (as reported by Fox et al., 2014; Hester et al., 2019; Nogueira et al.,
2022). The sizes of the surface water-groundwater mixing zone (e.g., thickness and area) occupy a
small proportion of the whole HZ. In the immobile bedform, the thin mixing zones (16 ~ 84%
ranges) occupying ~10% is consistent with prior work (Hester et al., 2013; Santizo et al., 2020).
During the initiation of bedform migration, however, the mixing pattern, size, and intensity of
surface water-groundwater interactions undergo modification. A continuous SW-GW mixing zone is
formed within the ripples of the medium- to fast-moving bedform (Figure 4), and the area of mixing
zone increases to approximately 15 ~ 25% at this time. Besides, the net flux of surface water and
groundwater mixing across the sediment-water interface (or within the riverbed) is also increased
with stream velocity and bedform migration celerity (Figure 5). As a result, bedform migration
controls and determines the hotspots and magnitude of the SW and GW mixing. The bedforms are
typically assumed to be immobile potentially making underestimations of surface water and





groundwater mixing flux and mixing zone in a HZ.
Instead of the typical crescent-shaped mixing-dependent (MD) denitrifying zone observed in
stationary bedforms (Naranjo et al., 2015; Hester et al., 2014, 2019), the MD reaction zonation
changes to the layer shape distributed at the fringe of the HEF cells, where mixing between SW and
GW develop to a largest degree (Figure 5). Such a situation exists where the stream flows into with a
relative homogeneous sandy riverbed with low autochthonous organic carbon content and encounters
with nitrate enrich groundwater. The heterogeneous streambeds including buried autochthonous
organic matter (Sawyer, 2015; Ping et al., 2022), deposited particulate organic particles (Drummond
et al., 2017; Ping et al., 2023), and biological aggregate (Xian et al., 2022), would complicate the
hyporheic exchange process, induce the rough and irregular shapes and boundaries of HEF cells and
mixing zone, and therefore affect redox microenvironments and biogeochemical zonations. MD
denitrifying hotspot would also occur around available sources of DOC.

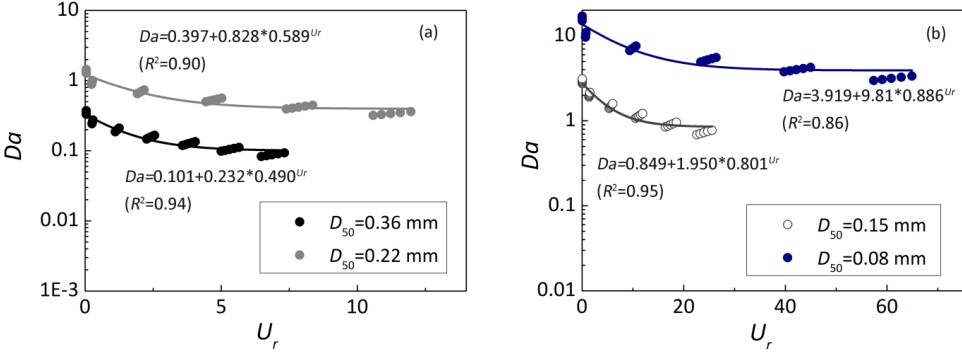

**Figure 10.** Variation of the dimensionless Damköhler number as a function of the dimensionless
parameter $U_r$.
Previous studies have demonstrated that migrating bedforms constrains the penetration depths
of stream-borne solute and reduces the removal efficiency of stream-borne nitrate (Jiang et al., 2022;



Kessler et al., 2015; Ping et al., 2022; Zheng et al., 2019). Our results revealed that the bedform
migration also reduces the HZ attenuation ability for groundwater borne nitrate. However, the
reasons for this phenomenon are different for various riverbed sediments. For very fine sand ($D_{50}$ =
0.08 mm), a larger celerity results in a decline in transport timescale with less impact on removal
efficiency as the system is rate-limited ($Da \gg 1$). For fine to medium sand riverbed ($D_{50}$ = 0.15–0.36
mm), the transport-limited situation leads to a low denitrification rate with increasing $U_r$ but
constantly decreasing $Da$ ($Da < 1$). It is important to note that the reaction timescale we calculated is
based on the consumption period of oxygen to a prescribed anoxic threshold. Under these conditions,
the exhaustion of labile DOC would also lead to the cessation of denitrification (Zarnetske et al.,
2011a, 2011b). These results demonstrate that in order to evaluate the self-purification capacity of the
HZ and its function as a natural barrier mitigating groundwater contamination, riverbed sediment
transport dynamics and grain size distributions need to be considered. Stabilizing bedform
configurations in restoration projects would enhances the natural attenuation capacity of the HZ.

Different from previous studies that demonstrated the magnitude of NMD denitrification was

often greater than that of MD denitrification (Hester et al., 2014; Trauth and Fleckenstein, 2017). The
results in our study show that the total reaction rate of s-$NO_3^-$ was smaller than that of g-$NO_3^-$ in
mobile bedforms. This phenomenon can be attributed to the following two reasons: First, the
concentration of s-$NO_3^-$ is one-third that of g-$NO_3^-$; Second, the reaction zone is reduced by
migration celerity for NMD denitrification, while the reaction zone for MD denitrification is
increased in the riverbed at the onset of bedform migration. The mixing intensity increases with
bedform migration, which facilitates the MD denitrification more effectively (Hester et al., 2019;
Nogueira et al., 2024; Trauth and Fleckenstein., 2017). Moreover, migration celerity increases the



influx of nitrate into the HZ by enhanced hyporheic exchange, while it has less impact on upwelling
GW flux. Hence, the removal efficiency of g-NO$_3^-$ is also higher than that of s-NO$_3^-$ in these
scenarios. More attention should be paid to the mixing dynamics and mixing triggered
biogeochemical reactions, which is helpful to put forward appropriate stream restoration plans so as
to enhance the health of the aquatic ecosystem (Hester et al., 2017; Lawrence et al., 2013).

In this study, we focused on ripples and, more broadly, shorter-wavelength topographic

roughness elements that form under low subcritical flow conditions in sandy riverbeds (Ashley,
1990; Gomez-Velez et al., 2015; Raudkivi, 1997). The undulating bedforms maintain dynamic
equilibrium through geometric adjustments, with their geometry remaining unchanged as the stream
velocity fluctuates within a specific range (10–30 cm/s). When stream velocities exceed this upper
threshold, a condition commonly observed in fast-flowing rivers, bedform geometries can be altered,
ultimately leading to bedform erosion (Boano et al., 2013; Harvey et al., 2012). This process is not
accounted for in the current model. If small-scale ripples develop and merge with larger-scale ripples
and dunes under moderate stream velocities, the removal efficiency of s-NO$_3^-$ and g-NO$_3^-$ may be
enhanced due to the extended hyporheic flow paths and increased residence timescales (Harvey et
al., 2012; Zomer and Hoitink, 2024). Otherwise, the removal of s-NO$_3^-$ would be highly hindered
because of shorter residence time and fully oxic condition in fast moving bedforms and fast flowing
rivers, while the removal of g-NO$_3^-$ would likely be less affected within the immobile streambeds.

In our model, dissimilatory nitrate reduction to ammonium (DNRA) was not incorporated,

given that denitrification is typically regarded as the predominant pathway for nitrate removal,
whereas DNRA plays a secondary role in nitrate transformation (Zarnetske et al., 2012). Lansdown et
al. (2012) and Quick et al. (2016) have demonstrated that approximately 5% of $^{15}$NO$_3^-$ tracer in river



sediment incubations underwent DNRA, while 85% underwent denitrification. Nevertheless, DNRA
competes with denitrification for $NO_3^-$ and DOC as electron acceptor and donors within HZs. When
an oligotrophic and/or a pristine stream infiltrate into the streambed and subsequently interact and
mix with nitrate-enriched groundwater, the MD DNRA would not occur due to the low C/N ratio.
MD denitrification zone shifts upstream toward the overlying water column, leading to a pronounced
spatial mismatch between the denitrification zone and the mixing interface. This occurs because
DOC is intensively consumed within the HEF cell, such that elevated MD denitrification rates
emerge below the DOC and oxygen plumes yet above the mixing zone. Specifically, this
phenomenon can be attributed to the critical role of dispersion effect in solute transport and mixing-
triggered denitrification, besides advection effect. When an eutrophication stream with higher DOC
concentration, DNRA would have a greater influence in nitrate transformation because DNRA is
prone to occur in $NO_3^-$-limited (that is DOC sufficient) conditions compared to denitrification (Zhu
et al., 2023). The ammonia produced by mixing-dependent DNRA would be further nitrified within
the aerobic HEF cell, thereby potentially elevating the risk of nitrate pollution in surface water.

In our model, stream velocity and upward groundwater flux are considered constant in the

present model, yet they may change in time due to storm events, tidal pumping, snowmelt, or
reservoir hydro-peaking (Liu et al., 2024; Nogueira et al., 2022; Song et al., 2018). Hester et al.
(2019) demonstrated that increasing surface water stage would enhance both NMD and MD
denitrification. Nogueira et al. (2024) and Trauth and Fleckenstein (2017) pointed out that
groundwater discharge events increase the magnitude of surface water-groundwater mixing,
therefore effecting the prevalence of MD denitrification. The interactions among morphological
dynamics, hyporheic exchange, and biogeochemical processes under transient conditions are key



areas for future research.
**5. Conclusion**

The numerical model developed in this study was applied to simulate the interaction and mixing

of upwelling groundwater with bedform-induced hyporheic flow, examining how bedform migration
influences surface water and groundwater mixing and the processing of groundwater-borne nitrate
within the HZ. Our analysis quantified the mixing flux and the size of mixing zone, as well as the
mixing-dependent denitrification rates and removal efficiencies across riverbed sediments
characterized by varying grain sizes, stream flow velocities, and groundwater discharge fluxes. These
model simulations reveal that as bedforms migrate, the surface water-groundwater mixing zone and
the associated mixing-dependent denitrification zone progressively evolve into uniform, band-like
structures. For riverbeds composed of fine to medium sand ($D_{50}$ = 0.15–0.36 mm), both the
magnitudes of SW and GW exchange flux and mixing flux increase significantly when turnover
becomes the dominant exchange mechanism, while the proportion of mixing flux across the
sediment-water interface and the size of mixing zone remain approximately constant at this time.
Meanwhile, both the mixing-dependent denitrification rates and removal efficiencies decline
significantly with increasing stream flow velocities and associated bedform migration rates. Under
dynamic bedform conditions, the self-purification capacity of the HZ is reduced, compromising its
role as a natural barrier against groundwater contamination. The management of aquatic systems
involving riverbed sediments can be enhanced by incorporating the analyzed factors identified here,
particularly when management goals encompass the removal of groundwater borne nitrate.



**Notation**

| | |
|---|---|
| $S$ | Stream slope [-] |
| $H$ | Water depth [L] |
| $U$ | Stream velocity [L T$^{-1}$] |
| $\lambda$ | Ripple wavelength [L] |
| $u_c$ | Bedform migration celerity [L T$^{-1}$] |
| $l$ | Streambed height [L] |
| $H_d$ | Ripple height [L] |
| $D_{50}$ | Median sediment size [L] |
| $x$ | Horizontal coordinate, rightward positive [-] |
| $y$ | Vertical coordinate, upward positive [-] |
| $k$ | Sediment permeability [L$^2$] |
| $K$ | Sediment hydraulic conductivity [L T$^{-1}$] |
| $h$ | Hydraulic head [L] |
| $h_m$ | Amplitude of the sinusoidal head variation [L] |
| $m$ | Wavenumber of the variation [-] |
| $g$ | Gravity acceleration [L T$^{-2}$] |
| $c_i$ | Concentration of reactive components [M L$^{-3}$] |
| $c_{gw}$ | Groundwater tracer [M L$^{-3}$] |
| $v$ | Seepage velocity [L T$^{-1}$] |
| $\theta$ | Sediment porosity [-] |
| $\alpha_L$ | Longitudinal dispersivities |





| $\alpha_T$ | Transverse dispersivities [L] |
|---|---|
| $\boldsymbol{D_{ij}}$ | Hydrodynamic dispersion [$L^2\,T^{-1}$] |
| $D_m$ | Molecular diffusion coefficient [$L^2\,T^{-1}$] |
| $\iota$ | Tortuosity factor [-] |
| $V_{AR}$ | Maximum reaction rate of aerobic respiration [$T^{-1}$] |
| $V_{DN}$ | Maximum reaction rate of denitrification [$T^{-1}$] |
| $K_{inh}$ | Non-competitive inhibition factor [$M\,L^{-3}$] |
| $K_{DOC}$ | Half-saturation for dissolved organic carbon [$M\,L^{-3}$] |
| $K_{NO3^-}$ | Half-saturation for nitrate [$M\,L^{-3}$] |
| $K_{O2}$ | Half-saturation for oxygen [$M\,L^{-3}$] |
| $X_{AR}$ | Microbial concentration facilitating aerobic respiration [$M\,L^{-3}$] |
| $X_{DN}$ | Microbial concentration facilitating denitrification [$M\,L^{-3}$] |
| $\rho_s$ | Sediment density [$M\,L^{-3}$] |
| $\rho$ | Water density [$M\,L^{-3}$] |
| $P$ | Hydraulic pressure [$M\,L^{-1}T^{-2}$] |
| $u^*$ | Bed shear velocity [$L\,T^{-1}$] |
| $u_{cr}^*$ | Critical bed shear velocity [$L\,T^{-1}$] |
| $n$ | Manning coefficient [-] |
| $\tau^*$ | Shield parameter [-] |
| $\tau_{cr}$ | Critical shear stress [$M\,L^{-1}T^{-2}$] |
| $\tau_{cr}^*$ | Critical Shield parameter [-] |
| $r$ | Submerged specific gravity of sediment [-] |



| $R_{O2}$ | Aerobic respiration rate [M L$^{-3}$T$^{-1}$] |
|---|---|
| $R_{s-NO3}$ | Non-mixing-dependent denitrification rate [M L$^{-3}$T$^{-1}$] |
| $R_{g-NO3}$ | Mixing-dependent denitrification rate [M L$^{-3}$T$^{-1}$] |
| $R_{DOC}$ | Dissolved oxygen carbon consumption rate [M L$^{-3}$T$^{-1}$] |
| $\upsilon$ | Kinematic viscosity of water [L$^2$ T$^{-1}$] |
| $u_s$ | Underflow seepage velocity induced by stream gradient [L T$^{-1}$] |
| $Re$ | Reynolds number [-] |
| $U_r$ | ratio of bedform celerity to pore water velocity [-] |
| $u_p$ | Darcy velocity induced by pumping process [L T$^{-1}$] |
| $u_q$ | Vertical groundwater flux [L T$^{-1}$] |
| $U_b$ | Ratio of vertical groundwater flux to hyporheic exchange flux |
| $\tau_R$ | Biogeochemical reaction timescale [T] |
| $\tau_T$ | Water transport timescale [T] |
| $Da$ | Damköhler number [-] |
| $\Delta x$ | Bedform migrating displacement per timestep (L) |
| $dt$ | The length of per timestep (T) |
| $Da'$ | Conversion factor for unit Darcy to m$^2$ [-] |
| $F_{mix}$ | the proportion of mixing flux to hyporheic exchange flux [-] |
| $A_{mix}$ | Surface water and groundwater mixing zone [L$^2$] |
| $A$ | Streambed area [L$^2$] |
| $M_{NDN}$ | Nitrate removed by non-mixing dependent denitrification [M] |
| $M_{DN}$ | Nitrate removed by mixing dependent denitrification [M] |



| $N_{RE}$ | Nitrate removal efficiency [-] |
|---|---|
| $M_{in}$ | Nitrate being introduced into streambed [M] |

Abbreviations

| HZ | Hyporheic zone |
|---|---|
| SW | Surface water |
| GW | Groundwater |
| HEF | Hyporheic exchange flow |
| MF | Mixing flux |
| NMD | Non-mixing dependent |
| MD | Mixing dependent |
| AR | Aerobic respiration |
| DN | Denitrification |

**Supporting information**
Additional details of the model scenarios and model validation were displayed in the supporting
information.
**Data availability**
All raw data can be provided by the first author upon request.
**Competing interest**
The authors declare that they have no conflict of interest.
**Author contribution**



Conceptualization: XP, YX
Formal analysis: XP
Funding acquisition: XP, ZW, YX, SK
Investigation: XP, ZW
Methodology: XP, ZW, YX
Writing-original draft: XP
Writing-review and editing: ZW, YX, MJ, SK
Project administration: ZW
**Acknowledgments**
The work was financially supported by the National Natural Science Foundation of China (Nos.
U23A2042, 42407075 and 42107089). This study was also supported by the "CUG Scholar"
Scientific Research Funds at China University of Geosciences (Wuhan) (Project No.2023067), and
supported by the Postdoctoral Fellowship Program of China Postdoctoral Science Foundation under
Grant Number GZC20241601. SK has been supported by the Royal Society (INF\R2\212060).

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
