# Peer review of "Modeling surface water and groundwater mixing and mixing-dependent denitrification"

_EGUsphere, 2025_

## Author Comment (AC4)

General comments: This study investigates the effects of migrating bedforms on surface-groundwater mixing and denitrification in a gaining stream reach. The authors employ a numerical model that couples Darcy flow with a translating sinusoidal head boundary and a multi-species reactive transport model. While the paper introduces new metrics such as "mixing flux," "mixing fraction," and "mixing-zone area," my primary concern is the substantial overlap with the authors' previous work (2022). The modeling framework, numerical implementation, and validation appear nearly identical.

The claimed novelty rests on applying the existing model to a different scenario and introducing new diagnostic metrics. However, the justification for these metrics is insufficient, and their introduction does not lead to the discovery of new physical mechanisms. Furthermore, the main conclusion: bedform migration shortens residence times and thereby reduces denitrification—reiterates, is a key finding from the 2022 paper (https://doi.org/10.1029/2022WR033258). The manuscript reads more as an incremental application of prior work rather than a standalone scientific article with a distinct and significant contribution.

Response to the general comments: We thank this reviewer for the constructive comments on our manuscript. We think the main concern of this reviewer is the difference between this study and our previous publication (Ping et al., 2022). We will classify the difference and the novelty of this study clearly in the revised version. Actually, this study is quite different from Ping et al. (2022) because: 1) Firstly, in this study, we focus on downstream section of rivers under gaining conditions firstly to simulate surface water-groundwater mixing processes under dynamic bedforms, while Ping et al. (2022) focused on hyporheic exchange processes in the headwater to midstream sections of rivers. therefore, the research focus is quite different for these two studies. 2) Secondly, the mixing process has been considered in this study. As far as we know, this process has not been thoroughly investigated in the migrating ripple model (Jiang et al., 2022; Zheng et al., 2019; Kessler et al., 2015). It is notable that the bedform migration affect mixing process significantly. 3) Thirdly, Ping et al. (2022) only considered stream-borne solutes and non-mixing dependent reactions, while in this study we evaluate and quantify the self-purification ability of the hyporheic zone for groundwater-borne contaminant. Therefore, we believe that this study is quite different from Ping et al. (2022). In addition, we have obtained several interesting findings in this study, which can be summarized as: 1)

bedform migration reshape the surface water-groundwater mixing zone and mixing-trigged biogeochemical zone patterns; 2) at slow to moderate migration rates, bedform migration expands the surface water-groundwater mixing zone and increases the fraction of mixing in hyporheic flux, with both parameters stabilizing at a plateau thereafter; 3) a large bedform migration celerity limits mixing-dependent denitrification rate, weakening the hyporheic zone's role as a natural barrier against groundwater contaminant.

In summary, we appreciate the concern from this reviewer on the novelty of this manuscript. We will revise the paper carefully to make it clear on the novelty of this paper, and clearly state the difference between this study and Ping et al. (2022). We are planning to revise literature review, the schematic diagram (figure 1), and some other parts, if necessary, to make it clearer. We are pretty sure that this concern can be addressed and this paper can be considered for potential publication in HESS after revision.

1.The literature review should be expanded to include a more thorough discussion of numerical model development in this field. Critically, this section must clearly and explicitly state how this study's model and objectives differ from and advance upon the 2022 paper. The work appears to be a case study applying a previously established model. It does not introduce a new mechanistic framework, a dimensionless parameter space map, or any analytical/scaling relationships that would constitute an independent and generalizable scientific contribution.

Response to comment #1: We agree with this reviewer on this point. In the revised manuscript, we will make targeted revisions in Introduction section to thoroughly discuss the development of numerical models in this field, explicitly state the difference from the 2022 paper, and highlight the novelty of this research.

2.Thresholds used to define the mixing zone (e.g., 16%–84% concentration range) are presented without a clear physical or statistical basis. The robustness of the results is questionable without a sensitivity analysis of these thresholds, as well as the numerical dispersion and grid resolution. Furthermore, the newly proposed metrics are not validated against any established, well-accepted measures of mixing from the fluid dynamics or hydrology literature, making their utility and interpretation difficult to assess.

Response to comment #2: We will supplement targeted analyses and validation to address these concerns, and we elaborate on the revisions and scientific rationale as follows:

1) Actually, there is no standard for the threshold of the mixing zone; however, it generally accepted that the mixing zone ranges from 10% to 90% in terms of groundwater proportion (Hester et al., 2013; Woessner et al., 2000). This interval effectively distinguishes the mixing zone, where groundwater and surface water interact dynamically, from the two endmembers: pure groundwater (>90%) and pure surface water (<10%). In this study, we selected the 16–84% range, which corresponds to the ±1σ interval of a normal distribution, as proposed by Santizo et al. (2020). To assess how threshold variations influence results, we will perform a sensitivity analysis that tests three alternative concentration ranges: 10%–90% (wider interval), 16%–84%, and 20%–80% (narrower range) in the revised paper.

2) We agree that the numerical dispersion and grid resolution can influence solute transport and mixing results. We will test three grid sizes to assess spatial discretization impacts: baseline grid (the original resolution), fine grid and coarse grid. Critical parameters would be compared for each grid size, confirming that the baseline grid is sufficiently resolved to capture mixing dynamics and minimize numerical dispersion.

3) The calculations of mixing metrics in this study follow the methods outlined by Hester et al. (2013, 2014). These metrics are not newly proposed by this study; instead, they have been widely applied and validated in previous studies (Santizo et al., 2020, 2022; Nogueira et al., 2022; Woessner et al., 2000), the evidence that supports their rationality. We will state that clearly in the revised manuscript.

3.The model's idealizations limit the generalizability of its findings. Key simplifications include: (a) representing bedform migration as a simple translating sinusoidal head, which neglects morphodynamic feedbacks; (b) using a 2-D domain with a single bedform wavelength and amplitude; and (c) omitting sediment heterogeneity, such as grain size variations or layering, which are known to strongly influence hyporheic exchange.

Response to comment #3: (a) Actually, simplifying undulating riverbeds to a horizontal configuration and applying a sinusoidal head distribution at the sediment-water interface exerts minimal impacts on both the hyporheic exchange flux (HEF) driven by the riverbed and solute transport within the

riverbed, which has been proven by many researchers (Elliott & Brooks, 1997a, b; Eylers, 1994; Rutherford et al., 1995). Indeed, we have conducted an additional simulation incorporating ripple geometry and compared it to the flat-bed model: both yield similar HEF and solute plume characteristics (this simulation will be added to the revised manuscript).

(b) Given the periodicity of bedform geometry in our model, we defined the model domain using a single bedform wavelength and amplitude. This is a well-established practice in modeling, as it minimizes edge effects without the computational intensity of full-scale simulations of multiple consecutive bedforms. To validate this simplification, we simulated three consecutive ripples, focusing on the middle one: comparisons of its pressure, solute concentration, and vertical boundary gradients with the single ripple model showed only minor differences. We will state that the model applies most directly to straight, low-curvature streams with uniform bedform spacing (typical of agricultural/urban downstream reaches, consistent with our focus; Hester et al., 2014; Krause et al., 2022). Here, hyporheic exchange is dominated by streamwise-vertical flow cells, with lateral hyporheic flux contributing little to total exchange.

(c) We omitted this factor in the current study to isolating the effects of bedform migration on surface water-groundwater mixing and mixing-triggered reaction. This work specifically targets downstream gaining reaches characterized by fine sand beds, which are a relatively homogeneous sediment type. To address this limitation, we will add relevant context to the Discussion section of the revised manuscript. We will propose that future studies integrate stochastic K fields to explore how sediment heterogeneity interacts with bedform migration, for instance, whether high-K hotspots amplify or dampen migration-driven mixing.